# Physics-aware Difference Graph Networks for Sparsely-Observed Dynamics

**Sungyong Seo,**[*] **Chuizheng Meng,**[*] **Yan Liu**
Department of Computer Science
University of Southern California
`{sungyons,chuizhem,yanliu.cs}@usc.edu`

## Abstract

Sparsely available data points cause numerical error on finite differences which hinders us from modeling the dynamics of physical systems. The discretization error becomes even larger when the sparse data are irregularly distributed or defined on an unstructured grid, making it hard to build deep learning models to handle physics-governing observations on the unstructured grid. In this paper, we propose a novel architecture, Physics-aware Difference Graph Networks (PA-DGN), which exploits neighboring information to learn finite differences inspired by physics equations. PA-DGN leverages data-driven end-to-end learning to discover underlying dynamical relations between the spatial and temporal differences in given sequential observations. We demonstrate the superiority of PA-DGN in the approximation of directional derivatives and the prediction of graph signals on the synthetic data and the real-world climate observations from weather stations.

## 1 Introduction

Modeling real world phenomena, such as climate observations, traffic flow, physics and chemistry simulation (Li et al., 2018; Geng et al., 2019; Long et al., 2018; de Bezenac et al., 2018; Sanchez-Gonzalez et al., 2018; Gilmer et al., 2017), is important but extremely challenging. While deep learning has achieved remarkable successes in prediction tasks by learning latent representations from *data-rich* applications such as image recognition (Krizhevsky et al., 2012), text understanding (Wu et al., 2016), and speech recognition (Hinton et al., 2012), we confront many challenging scenarios in modeling natural phenomena with deep neural networks when only a limited number of observations are available. Particularly, the sparsely available data points cause substantial numerical error when we utilize existing finite difference operators and the limitation requires a more principled way to redesign deep learning models.

While many methods have been proposed to model physics-simulated observations using deep learning, many of them are designed under the assumption that input is on a continuous domain. For example, Raissi et al. (2017a;b) proposed physics-informed neural networks (PINNs) to learn nonlinear relations between input (spatial- and temporal-coordinates $(x, t)$) and output simulated with a given partial differential equation (PDE). Since Raissi et al. (2017a;b) use the coordinates as input and compute derivatives based on the coordinates to represent the equation, the setting is only valid when the data are densely observed over spatial and temporal space.

Prior knowledge related to physics equations has been combined with data-driven models for various purposes. Chen et al. (2015) proposed a nonlinear diffusion process for image restoration and de Bezenac et al. (2018) incorporated the transport physics (advection-diffusion equation) with deep neural networks for forecasting sea surface temperature by extracting the motion field. Lutter et al. (2019) introduced deep Lagrangian networks specialized to learn Lagrangian mechanics with learnable parameters. Seo & Liu (2019) proposed a physics-informed regularizer to impose data-specific physics equations. In common, the methods in Chen et al. (2015); de Bezenac et al. (2018); Lutter et al. (2019) are not efficiently applicable to sparsely discretized input as only a small number of data points are available and continuous properties on given space are not easily recovered. It is

---

[*]Equally contributed.

unsuitable to directly use continuous differential operators to provide local behaviors because it is hard to approximate the continuous derivatives precisely with the sparse points (Shewchuk, 2002; Amenta & Kil, 2004; Luo et al., 2009). Furthermore, they are only applicable when the specific physics equations are explicitly given and still hard to be generalized to incorporate other types of equations.

As another direction to modeling physics-simulated data, Long et al. (2018) proposed PDE-Net which uncovers the underlying hidden PDEs and predicts the dynamics of complex systems. Ruthotto & Haber (2018) derived new CNNs: parabolic and hyperbolic CNNs based on ResNet (He et al., 2016) architecture motivated by PDE theory. While Long et al. (2018); Ruthotto & Haber (2018) are flexible to uncover hidden physics from the constrained kernels, it is still restrictive to a regular grid where the proposed constraints on the learnable filters are easily defined.

The topic of reasoning physical dynamics of discrete objects has been actively studied (Sanchez-Gonzalez et al., 2018; Battaglia et al., 2016; Chang et al., 2016) as the appearance of graph-based neural networks (Kipf & Welling, 2017; Santoro et al., 2017; Gilmer et al., 2017). Although these models can handle sparsely located data points without explicitly given physics equations, they are purely data-driven so that the physics-inspired inductive bias for exploiting *finite differences* is not considered at all. In contrast, our method consists of physics-aware modules allowing efficiently leveraging the inductive bias to learn spatiotemporal data from the physics system.

In this paper, we propose physics-aware difference graph networks (PA-DGN) whose architecture is motivated to leverage *differences* of sparsely available data from the physical systems. The *differences* are particularly important since most of the physics-related dynamic equations (e.g., Navier–Stokes equations) handle differences of physical quantities in spatial and temporal space instead of using the quantities directly. Inspired by the property, we first propose spatial difference layer (SDL) to efficiently learn the local representations by aggregating neighboring information in the sparse data points. The layer is based on graph networks (GN) as it easily leverages structural features to learn the localized representations and share the parameters for computing the localized features. Then, the layer is followed by recurrent graph networks (RGN) to predict the temporal difference which is another core component of physics-related dynamic equations. PA-DGN is applicable to various tasks and we provide two representative tasks; the approximation of directional derivatives and the prediction of graph signals.

Our contributions are:

- We tackle a limitation of the sparsely discretized data which cause numerical error when we model the physical system by proposing spatial difference layer (SDL) for efficiently exploiting neighboring information under the limitation of sparsely observable points.
- We combine SDL with recurrent graph networks to build PA-DGN which automatically learns the underlying spatiotemporal dynamics in graph signals.
- We verify that PA-DGN is effective in approximating directional derivatives and predicting graph signals in synthetic data. Then, we conduct exhaustive experiments to predict climate observations from land-based weather stations and demonstrate that PA-DGN outperforms other baselines.

## 2 PHYSICS-AWARE DIFFERENCE GRAPH NETWORK

In this section, we introduce the building module used to learn spatial differences of graph signals and describe how the module is used to predict signals in the physics system.

### 2.1 DIFFERENCE OPERATORS ON GRAPH

As approximation of derivatives in continuous domain, difference operators have been used as a core role to compute numerical solutions of (continuous) differential equations. Since it is hard to derive closed-form expressions of derivatives in real-world data, the difference operators have been considered as alternative tools to describe and solve PDEs in practice. The operators are especially important for physics-related data (e.g., meteorological observations) because the governing rules behind the observations are mostly differential equations.

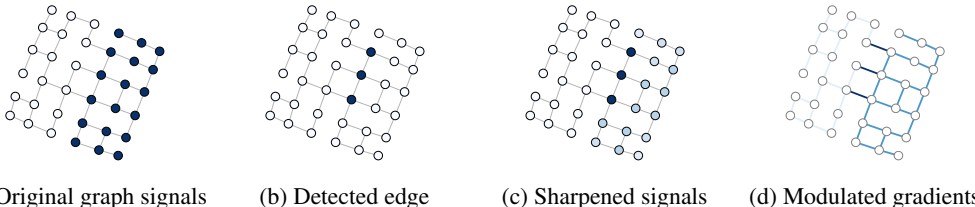

(a) Original graph signals     (b) Detected edge     (c) Sharpened signals     (d) Modulated gradients

Figure 1: Examples of difference operators applied to graph signal. Filters used for the processing are (b) $\sum_j (f_i - f_j)$ (c) $\sum_j (1.1 f_i - f_j)$, (d) $f_j - 0.5 f_i$.

**Graph signals**  Given a graph $\mathcal{G} = (\mathbb{V}, \mathbb{E})$ where $\mathbb{V}$ is a set of vertices $\mathbb{V} = \{1, \ldots, N_v\}$ and $\mathbb{E}$ is a set of edges $\mathbb{E} \subseteq \{(i,j) | i, j \in \mathbb{V}\}$ ($|\mathbb{E}| = N_e$), graph signals on all nodes at time $t$ are $f(t) = \{f_i(t) \mid i \in \mathbb{V}\}$ where $f_i : \mathbb{V} \to \mathbb{R}$. Graph signals on edges can also be defined similarly, $F(t) = \{F_{ij}(t) \mid (i,j) \in \mathbb{E}\}$ where $F_{ij} : \mathbb{E} \to \mathbb{R}$. Both signals can be multidimensional.

**Gradient on graph**  The *gradient* ($\nabla$) of a function on nodes of a graph is represented by finite difference

$$\nabla : L^2(\mathbb{V}) \to L^2(\mathbb{E}), \qquad (\nabla f)_{ij} = (f_j - f_i) \quad \text{if } (i,j) \in \mathbb{E} \text{ and } 0 \text{ otherwise,}$$

where $L^2(\mathbb{V})$ and $L^2(\mathbb{E})$ denote Hilbert spaces for node/edge functions, respectively. The gradients on a graph provide *finite differences* of graph signals and they become edge $(i,j)$ features.

**Laplace-Beltrami operator**  *Laplace-Beltrami* operator (or Laplacian, $\Delta$) in graph domain is defined as

$$\Delta : L^2(\mathbb{V}) \to L^2(\mathbb{V}), \qquad (\Delta f)_i = \sum_{j:(i,j) \in \mathbb{E}} (f_i - f_j) \quad \forall i, j \in \mathbb{V}$$

This operator is usually regarded as a matrix form in other literature, $\boldsymbol{L} = \boldsymbol{D} - \boldsymbol{A}$ where $\boldsymbol{A}$ is an adjacency matrix and $\boldsymbol{D} = \mathrm{diag}(\sum_{j:j \neq i} \boldsymbol{A}_{ij})$ is a degree matrix.

## 2.2 DIFFERENCE OPERATORS ON TRIANGULATED MESH

According to Crane (2018), the gradient and Laplacian operators on the triangulated mesh can be discretized by incorporating the coordinates of nodes. To obtain the gradient operator, the per-face gradient of each triangular face is calculated first. Then, the gradient on each node is the area-weighted average of all its neighboring faces, and the gradient on edge $(i,j)$ is defined as the dot product between the per-node gradient and the direction vector $\boldsymbol{e}_{ij}$. The Laplacian operator can be discretized with Finite Element Method (FEM):

$$(\Delta f)_i = \frac{1}{2} \sum_{j:(i,j) \in \mathbb{E}} (\cot \alpha_j + \cot \beta_j)(f_j - f_i),$$

where node $j$ belongs to node $i$'s immediate neighbors ($j \in \mathbb{N}_i$) and $(\alpha_j, \beta_j)$ are two opposing angles of the edge $(i,j)$.

## 2.3 SPATIAL DIFFERENCE LAYER

While the difference operators are generalized in Riemannian manifolds (Lai et al., 2013; Lim, 2015), there exist numerical error compared to those in continuous space and the error can be larger when the nodes are spatially far from neighboring nodes because the connected nodes ($j \in \mathbb{N}_i$) of $i$-th node fail to represent local features around the node. Furthermore, the error is even larger if available data points are sparsely distributed (e.g., sensor-based observations). In other words, the difference operators are unlikely to discover meaningful spatial variations behind the sparse observations since they are highly limited to immediate neighboring information only. To mitigate the limitation, we propose spatial difference layer (SDL) which consists of a set of parameters to define learnable

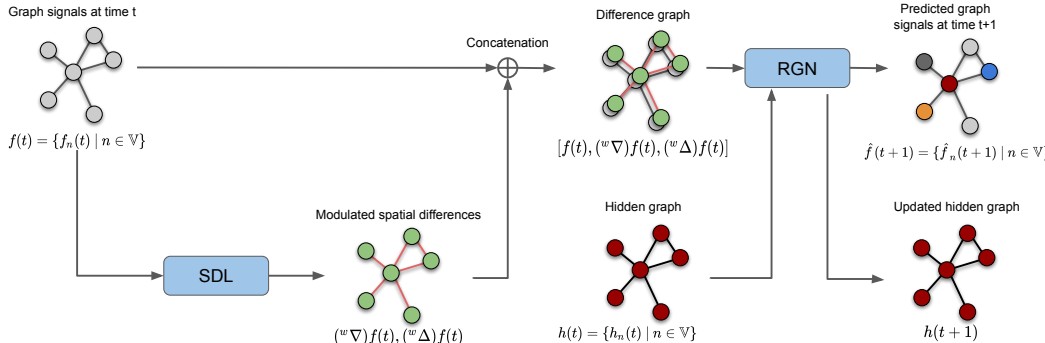

Figure 2: Physics-aware Difference Graph Networks for graph signal prediction. Blue boxes have learnable parameters and all parameters are trained through end-to-end learning. The nodes/edges can be multidimensional.

difference operators as a form of *gradient* and *Laplacian* to fully utilize neighboring information:

$$({}^w\nabla f)_{ij} = w_{ij}^{(g_1)}(f_j - w_{ij}^{(g_2)}f_i), \qquad ({}^w\Delta f)_i = \sum_{j:(i,j)\in\mathbb{E}} w_{ij}^{(l_1)}(f_i - w_{ij}^{(l_2)}f_j) \qquad (1)$$

where $w_{ij}$ are the parameters tuning the difference operators along with the corresponding edge direction $e_{ij}$. The two forms (Eq 1) are associated with edge and node features, respectively. The superscript in ${}^w\nabla$ and ${}^w\Delta$ denotes that the difference operators are functions of the learnable parameters $w$. $w_{ij}^{(g)}$ and $w_{ij}^{(l)}$ are obtained by integrating local information as follow:

$$w_{ij} = g(\{f_k, F_{mn} \mid k, (m,n) \in h\text{-hop neighborhood of edge } (i,j)\}) \qquad (2)$$

While the standard difference operators consider two connected nodes only ($i$ and $j$) for each edge $(i,j)$, Eq 2 uses a larger view ($h$-hop) to represent the differences between $i$ and $j$ nodes. Since graph networks (GN) (Battaglia et al., 2018) are efficient networks to aggregate neighboring information, we use GN for $g(\cdot)$ function and $w_{ij}$ are edge features of output of GN. Eq 2 can be viewed as a higher-order difference equation because nodes/edges which are multi-hop apart are considered.

$w_{ij}$ has a similar role of parameters in convolution kernels of CNNs. For example, while the standard gradient operator can be regarded as an example of simple edge-detecting filters, the operator can be a sharpening filter if $w_{ij}^{(g_1)} = 1$ and $w_{ij}^{(g_2)} = \frac{|\mathbb{N}_i|+1}{|\mathbb{N}_i|}$ for $i$ node and the operators over each edge are summed. In other words, by modulating $w_{ij}$, it is readily extended to conventional kernels including edge detection or sharpening filters and even further complicated kernels. On top of $w_{ij}$, the *difference* forms in Eq 1 make an optimizing process for learnable parameters based on the differences instead of the values intentionally. Eq 1 thus naturally provides the physics-inspired inductive bias which is particularly effective for modeling physics-related observations. Furthermore, it is easily possible to increase the number of channels for $w_{ij}^{(g)}$ and $w_{ij}^{(l)}$ to be more expressive. Figure 1 illustrates how the exemplary filters convolve the given graph signals.

## 2.4 RECURRENT GRAPH NETWORKS

**Difference graph**  Once the modulated spatial differences (${}^w\nabla f(t), {}^w\Delta f(t)$) are obtained, they are concatenated with the current signals $f(t)$ to construct node-wise ($z_i$) and edge-wise ($z_{ij}$) features and the graph is called a *difference graph*. The difference graph includes all information to describe spatial variations.

**Recurrent graph networks**  Given a snapshot ($f(t), F(t)$) of a sequence of graph signals, one difference graph is obtained and is used to predict next graph signals. While a non-linear layer can be used to combine the learned spatial differences to predict the next signals, it is limited to discover spatial relations only among the features in the difference graph. Since many equations describing physics-related phenomena are non-static (e.g., Navier–Stokes equations), we adopt recurrent graph networks (RGN) (Sanchez-Gonzalez et al., 2018) with a graph state $\mathcal{G}_h$ as input to combine the spatial

differences with temporal variations. RGN returns a graph state $(\mathcal{G}_h^* = (\boldsymbol{h}^{*(v)}, \boldsymbol{h}^{*(e)}))$ and next graph signal $\boldsymbol{z}_i^*$ and $\boldsymbol{z}_{ij}^*$. The update rule is described as follow:

1. $(\boldsymbol{z}_{ij}^*, \boldsymbol{h}^{*(e)}) \leftarrow \phi^e(\boldsymbol{z}_{ij}, \boldsymbol{z}_i, \boldsymbol{z}_j, \boldsymbol{h}^{(e)})$ for all $(i,j) \in \mathbb{E}$ pairs,

2. $(\boldsymbol{z}_i^*, \boldsymbol{h}^{*(v)}) \leftarrow \phi^v(\boldsymbol{z}_i, \bar{\boldsymbol{z}}_i', \boldsymbol{h}^{(v)})$ for all $i \in \mathbb{V}$,
   $\bar{z}_i'$ is an aggregated edge attribute related to the node $i$,

where $\phi^e, \phi^v$ are edge and node update functions, respectively, and they can be any recurrent unit. Finally, the prediction is made through a decoder by feeding the graph signal, $\boldsymbol{z}_i^*$ and $\boldsymbol{z}_{ij}^*$.

**Learning objective** Let $\hat{f}$ and $\hat{F}$ denote predictions of the target node/edge signals. PA-DGN is trained by minimizing the following objective:

$$\mathcal{L} = \sum_{i \in \mathbb{V}} ||f_i - \hat{f}_i||^2 + \sum_{(i,j) \in \mathbb{E}} ||F_{ij} - \hat{F}_{ij}||^2. \tag{3}$$

For multistep predictions, $\mathcal{L}$ is summed over all predicting steps. If only one type (node or edge) of signal is given, the corresponding term in Eq 3 is used to optimize the parameters in SDL and RGN simultaneously.

## 3 EFFECTIVENESS OF SPATIAL DIFFERENCE LAYER

To investigate if the proposed spatial difference forms (Eq 1) can be beneficial to learning physics-related patterns, we use SDL on two different tasks: (1) approximate directional derivatives and (2) predict synthetic graph signals.

### 3.1 APPROXIMATION OF DIRECTIONAL DERIVATIVES

As we claimed in Section 2.3, the standard difference forms (gradient and Laplacian) on a graph can cause significant numerical error easily because they are susceptible to a distance of two points and variations of a given function. To evaluate the applicability of the proposed SDL, we train SDL to approximate directional derivatives on a graph. First, we define a synthetic function and its gradients on 2D space and sample 200 points $(x_i, y_i)$. Then, we construct a graph on the sampled points by using $k$-NN algorithm ($k = 4$). With the known gradient $\left( \nabla f = (\frac{\partial f}{\partial x}, \frac{\partial f}{\partial y}) \right)$ at each point (a node in the graph), we can compute directional derivatives by projecting $\nabla f$ to a connected edge $\boldsymbol{e}_{ij}$ (See Figure 3). We compare against four baselines: (1) the finite gradient (**FinGrad**) (2) multilayer perceptron (**MLP**) (3) graph networks (**GN**) (4) a different form of Eq 1 (**One-w**). For the finite gradient $((f_j - f_i)/||\boldsymbol{x}_j - \boldsymbol{x}_i||)$, there is no learnable parameter and it only uses two points. For MLP, we feed $(f_i, f_j, \boldsymbol{x}_i, \boldsymbol{x}_j)$ as input to see whether learnable parameters can benefit the approximation or not. For GN, we use distances of two connected points as edge features and function values on the points as node features. The edge feature output of GN is used as a prediction for the directional derivative on the edge. Finally, we modify the proposed form as $(^w\nabla f)_{ij} = w_{ij}f_j - f_i$. GN and the modified form are used to verify the effectiveness of Eq 1. Note that we define two synthetic functions (Figure 4) which have different property; (1) monotonically increasing from a center and (2) periodically varying.

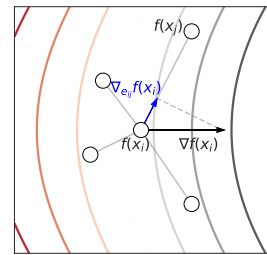

Figure 3: Directional derivative on graph

Table 1: Mean squared error ($10^{-2}$) for approximation of directional derivatives.

| Functions | FinGrad | MLP | GN | One-w | SDL |
|---|---|---|---|---|---|
| $f_1(x,y) = 0.1x^2 + 0.5y^2$ | 6.42±0.47 | 2.12±0.32 | 1.05±0.42 | 1.41±0.44 | **0.97**±0.39 |
| $f_2(x,y) = \sin(x) + \cos(y)$ | 5.90±0.04 | 2.29±0.77 | 2.17±0.34 | 6.73±1.17 | **1.26**±0.05 |

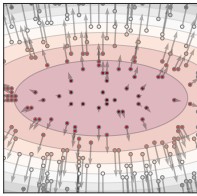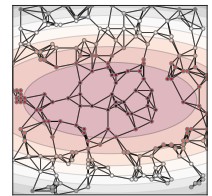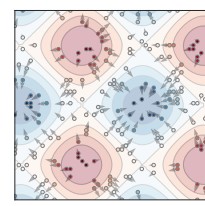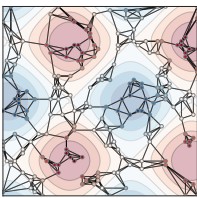

Figure 4: Gradients and graph structure of sampled points. Left: the synthetic function is $f_1(x, y) = 0.1x^2 + 0.5y^2$. Right: the synthetic function is $f_2(x, y) = \sin(x) + \cos(y)$.

**Approximation accuracy**    As shown in Table 1, the proposed spatial difference layer outperforms others by a large margin. As expected, FinGrad provides the largest error since it only considers two points without learnable parameters. It is found that the learnable parameters can significantly benefit to approximate the directional derivatives even if input is the same (FinGrad vs. MLP). Note that utilizing neighboring information (GN, One-w, SDL) is generally helpful to learn spatial variations properly. However, simply training parameters in GN is not sufficient and explicitly defining *difference*, which is important to understand spatial variations, provides more robust inductive bias. One important thing we found is that One-w is not effective as much as GN and it can be even worse than FinGrad. It is because of its limited degree of freedom. As implied in the form $(\nabla_w f)_{ij} = w_{ij} * f_j - f_i$, only one $w_{ij}$ adjusts the relative difference between $f_i$ and $f_j$, and this is not enough to learn whole possible linear combinations of $f_i$ and $f_j$. The unstable performance supports that the form of SDL is not ad-hoc but more rigorously designed.

## 3.2 GRAPH SIGNAL PREDICTION

We evaluate PA-DGN on the synthetic data sampled from the simulation of specific convection-diffusion equations, to provide if the proposed model can predict next signals of the simulated dynamics from observations on discrete nodes only. For the simulated dynamics, we use an equation slightly modified based on the one in Long et al. (2018).

$$\frac{df_i(t)}{dt} = a(i)(\nabla f)_{\hat{x}} + b(i)(\nabla f)_{\hat{y}} + c(i)\Delta f, \qquad f_i(0) = f_o(i), \qquad (4)$$

where the index $i$ points the $i$-th node whose coordinate is $(x_i, y_i)$ in the 2D space ($[0, 2\pi] \times [0, 2\pi]$) and $\hat{x}$ and $\hat{y}$ indicate $x$- and $y$-direction in the space. $a(i) = 0.5(\cos(y_i) + x_i(2\pi - x_i)\sin(x_i)) + 0.6$, $b(i) = 2(\cos(y_i) + \sin(x_i)) + 0.8$, and $c(i) = 0.5\left(1 - \frac{\sqrt{(x_i - \pi)^2 + (y_i - \pi)^2}}{\sqrt{2}\pi}\right)$. Then, we uniformly sample 250 points in the above 2D space. The task is to predict signal values of all points in the future $M$ steps given observed values of the first $N$ steps. For our experiments, we choose $N = 5$ and $M = 15$. Since there is no a priori graph structure on sampled points, we construct a graph with $k$-NN algorithm ($k = 4$) using the Euclidean distance. Figure 5 shows the dynamics and the graph structure of sampled points.

To evaluate the effect of the proposed SDL on the above prediction task, we cascade SDL and a linear regression model as our prediction model since the dynamics follows a linear partial differential equation. We compare its performance with four baselines: (1) vector auto-regressor (**VAR**); (2) multilayer perceptron (**MLP**); (3) **StandardOP**: the standard approximation of differential operators in Section 2.1 followed by a linear regressor; (4) **MeshOP**: similar to StandardOP but use the discretization on triangulated mesh in Section 2.2 for differential operators.

Table 2: Mean absolute error ($10^{-2}$) for graph signal prediction.

| VAR | MLP | StandardOP | MeshOP | SDL |
|---|---|---|---|---|
| 16.84±0.41 | 15.75±0.53 | 11.99±0.29 | 12.82±0.06 | **10.87**±0.98 |

**Prediction Performance**    Table 2 shows the prediction performance of different models measured with mean absolute error. The prediction model with our proposed spatial differential layer outperforms other baselines. All models incorporating any form of spatial difference operators (StandardOP,



$t = 1$  $t = 5$  $t = 10$  $t = 15$  $t = 20$

Figure 5: Synthetic dynamics and graph structure of sampled points.

MeshOP and SDL) outperform those without spatial difference operators (VAR and MLP), showing that introducing spatial differences information inspired by the intrinsic dynamics helps prediction. However, in cases where points with observable signal are sparse in the space, spatial difference operators derived with fixed rules can be inaccurate and sub-optimal for prediction since the locally linear assumption which they are based on no longer holds. Our proposed SDL, to the contrary, is capable of bridging the gap between approximated difference operators and accurate ones by introducing learnable coefficients utilizing neighboring information, and thus improves the prediction performance.

# 4 PREDICTION: GRAPH SIGNALS ON LAND-BASED WEATHER SENSORS

We evaluate the proposed model on the task of predicting climate observations (*Temperature*) from the land-based weather stations located in the United States.

## 4.1 EXPERIMENTAL SET-UP

**Data and task** We sample the weather stations located in the United States from the Online Climate Data Directory of the National Oceanic and Atmospheric Administration (NOAA) and choose the stations which have actively measured meteorological observations during 2015. We choose two geographically close but meteorologically diverse groups of stations: the Western and Southeastern states. We use $k$-Nearest Neighbor (NN) algorithm ($k = 4$) to generate graph structures and the final adjacency matrix is $A = (A_k + A_k^\top)/2$ to make it symmetric where $A_k$ is the output adjacency matrix from $k$-NN algorithm.

Figure 6 shows the distributions of the land-based weather stations and their connectivity. Since the stations are not synchronized and have different timestamps for the observations, we aggregate the time series hourly. The 1-year sequential data are split into the train set (8 months), the validation set (2 months), and the test set (2 months), respectively.

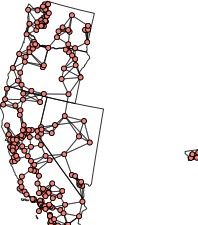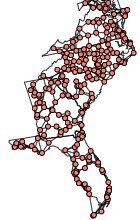

Our main task is to predict the next graph signals based on the current and past graph signals. All methods we evaluate are trained through the objective (Eq 3) with the Adam optimizer and we use scheduled sampling (Bengio et al., 2015) for the models with recurrent modules. We

Figure 6: Weather stations in (left) western (right) southeastern states in the United States and $k$-NN graph.

evaluate PA-DGN and other baselines on two prediction tasks, (1) 1-step and (2) multistep-ahead predictions. Furthermore, we demonstrate the ablation study that provides how much the spatial derivatives from our proposed SDL are important signals to predict the graph dynamics.

## 4.2 GRAPH SIGNAL PREDICTIONS

We compare against the widely used baselines (VAR, MLP, and GRU) for 1-step and multistep prediction. Then, we use RGN (Sanchez-Gonzalez et al., 2018) to examine how much the graph structure is beneficial. Finally, we evaluate PA-DGN to verify if the proposed architecture (Eq 1) is able to reduce prediction loss. Experiment results for the prediction task are summarized in Table 3.

Overall, RGN and PA-DGN are better than other baselines and it implies that the graph structure provides useful inductive bias for the task. It is intuitive as the meteorological observations are

continuously changing over the space and time and thus, the observations at the $i$-th station are strongly related to those of its neighboring stations.

PA-DGN outperforms RGN and the discrepancy comes from the fact that the spatial derivatives (Eq 1) we feed in PA-DGN are beneficial and this finding is expected because the meteorological signals at a certain point are a function of not only its previous signal but also the relative differences between neighbor signals and itself. Knowing the relative differences among local observations is particularly essential to understand physics-related dynamics. For example, Diffusion equation, which describes how physical quantities (e.g., heat) are transported through space over time, is also a function of relative differences of the quantities ($\frac{df}{dt} = D\Delta f$) rather than values of the neighbor signals. In other words, spatial differences are physics-aware features and it is desired to leverage the features as input to learn dynamics related to physical phenomena.

Table 3: Graph signal prediction results (MAE) on multistep predictions. In each row, we report the average with standard deviations from all baselines and PA-DGN. One step is 1-hour time interval.

| Region | Method | 1-step | 6-step | 12-step |
|--------|--------|--------|--------|---------|
| West | VAR | $0.1241 \pm 0.0234$ | $0.4295 \pm 0.1004$ | $0.4820 \pm 0.1298$ |
| | MLP | $0.1040 \pm 0.0003$ | $0.3742 \pm 0.0238$ | $0.4998 \pm 0.0637$ |
| | GRU | $0.0913 \pm 0.0047$ | $0.1871 \pm 0.0102$ | $0.2707 \pm 0.0006$ |
| | RGN | $0.0871 \pm 0.0033$ | $0.1708 \pm 0.0024$ | $0.2666 \pm 0.0252$ |
| | RGN(StandardOP) | $0.0860 \pm 0.0018$ | $0.1674 \pm 0.0019$ | $0.2504 \pm 0.0107$ |
| | RGN(MeshOP) | $0.0840 \pm 0.0015$ | $0.2119 \pm 0.0018$ | $0.4305 \pm 0.0177$ |
| | PA-DGN | $\mathbf{0.0840 \pm 0.0004}$ | $\mathbf{0.1614 \pm 0.0042}$ | $\mathbf{0.2439 \pm 0.0163}$ |
| SouthEast | VAR | $0.0889 \pm 0.0025$ | $0.2250 \pm 0.0013$ | $0.3062 \pm 0.0032$ |
| | MLP | $0.0722 \pm 0.0012$ | $0.1797 \pm 0.0086$ | $0.2514 \pm 0.0154$ |
| | GRU | $0.0751 \pm 0.0037$ | $0.1724 \pm 0.0130$ | $0.2446 \pm 0.0241$ |
| | RGN | $0.0790 \pm 0.0113$ | $0.1815 \pm 0.0239$ | $0.2548 \pm 0.0210$ |
| | RGN(StandardOP) | $0.0942 \pm 0.0121$ | $0.2135 \pm 0.0187$ | $0.2902 \pm 0.0348$ |
| | RGN(MeshOP) | $0.0905 \pm 0.0012$ | $0.2052 \pm 0.0012$ | $0.2602 \pm 0.0062$ |
| | PA-DGN | $\mathbf{0.0721 \pm 0.0002}$ | $\mathbf{0.1664 \pm 0.0011}$ | $\mathbf{0.2408 \pm 0.0056}$ |

## 4.3 CONTRIBUTION OF SPATIAL DERIVATIVES

We further investigate if the modulated spatial derivatives (Eq 1) are effectively advantageous compared to the spatial derivatives defined in Riemannian manifolds. First, RGN without any spatial derivatives is assessed for the prediction tasks on Western and Southeastern states graph signals. Note that this model does not use any extra features but the graph signal, $f(t)$. Secondly, we add (1) **StandardOP**, the discrete spatial differences (Gradient and Laplacian) in Section 2.1 and (2) **MeshOP**, the triangular mesh approximation of differential operators in Section 2.2 separately as additional signals to RGN. Finally, we incorporate with RGN our proposed Spatial Difference Layer.

Table 3 shows the contribution of each component. As expected, PA-DGN provides much higher drops in MAE (3.56%, 5.50%, 8.51% and 8.73%, 8.32%, 5.49% on two datasets, respectively) compared to RGN without derivatives and the results demonstrate that the derivatives, namely, relative differences from neighbor signals are effectively useful. However, neither RGN with StandardOP nor with MeshOP can consistently outperform RGN. We also found that PA-DGN consistently shows positive effects on the prediction error compared to the fixed derivatives. This finding is a piece of evidence to support that the parameters modulating spatial derivatives in our proposed Spacial Difference Layer are properly inferred to optimize the networks and to improve the prediction performance.

## 5 CONCLUSION

In this paper, we introduce a novel architecture (PA-DGN) that approximates spatial derivatives to use them to represent PDEs which have a prominent role for physics-aware modeling. PA-DGN

effectively learns the modulated derivatives for predictions and the derivatives can be used to discover hidden physics describing interactions between temporal and spatial derivatives.

ACKNOWLEDGEMENTS

This work is supported in part by NSF Research Grant IIS-1254206 and MINERVA grant N00014-17-1-2281, granted to co-author Yan Liu in her academic role at the University of Southern California. The views and conclusions are those of the authors and should not be interpreted as representing the official policies of the funding agency, or the U.S. Government. Last but not least, we appreciate anonymous reviewers for your thorough comments and suggestions.

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

## A    APPENDIX

### A.1    SIMULATED DATA

For the simulated dynamics, we discretize the following partial differential equation similar to the one in Long et al. (2018) to simulate the corresponding linear variable-coefficient convection-diffusion equation on graphs.

In a continuous space, we define the linear variable-coefficient convection-diffusion equation as:

$$\begin{cases} \frac{\partial f}{\partial t} & = a(x,y)f_x + b(x,y)f_y + c(x,y)\Delta f \\ f|_{t=0} & = f_0(x,y) \end{cases} \tag{5}$$

, with $\Omega = [0, 2\pi] \times [0, 2\pi]$, $(t, x, y) \in [0, 0.2] \times \Omega$, $a(x, y) = 0.5(\cos(y) + x(2\pi - x)\sin(x)) + 0.6$, $\quad b(x, y) = 2(\cos(y) + \sin(x)) + 0.8$, $c(x, y) = 0.5 \left( 1 - \frac{\sqrt{(x_i - \pi)^2 + (y_i - \pi)^2}}{\sqrt{2}\pi} \right)$.

We follow the setting of initialization in Long et al. (2018):

$$f_0(x, y) = \sum_{|k|,|l| \leq N} \lambda_{k,l} \cos(kx + ly) + \gamma_{k,l} \sin(kx + ly) \tag{6}$$

, where $N = 9$, $\lambda_{k,l}, \gamma_{k,l} \sim \mathcal{N}\left(0, \frac{1}{50}\right)$, and $k$ and $l$ are chosen randomly.

We use spatial difference operators to approximate spatial derivatives:

$$\begin{cases} f_x(x_i, y_i) & = \frac{1}{2s}(f(x_i, y_i) - f(x_i - s, y_i)) - \frac{1}{2s}(f(x_i, y_i) - f(x_i + s, y_i)) \\[2mm] f_y(x_i, y_i) & = \frac{1}{2s}(f(x_i, y_i) - f(x_i, y_i - s)) - \frac{1}{2s}(f(x_i, y_i) - f(x_i, y_i + s)) \\[2mm] f_{xx}(x_i, y_i) & = \frac{1}{s^2}(f(x_i, y_i) - f(x_i - s, y_i)) + \frac{1}{s^2}(f(x_i, y_i) - f(x_i + s, y_i)) \\[2mm] f_{yy}(x_i, y_i) & = \frac{1}{s^2}(f(x_i, y_i) - f(x_i, y_i - s)) + \frac{1}{s^2}(f(x_i, y_i) - f(x_i, y_i + s)) \end{cases} \tag{7}$$

, where $s$ is the spatial grid size for discretization.

Then we rewrite (5) with difference operators defined on graphs:

$$\begin{cases} \frac{\partial f}{\partial t} = a(i)(\nabla f)_{\hat{x}} + b(i)(\nabla f)_{\hat{y}} + c(i)((\Delta f)_{\hat{x}\hat{x}} + (\Delta f)_{\hat{y}\hat{y}}) \\ f_i(0) = f_o(i) \end{cases} \tag{8}$$

, where

$$a(i)(x_j, y_j) = \begin{cases} \dfrac{a(x_i, y_i)}{2s} & \text{if } x_i = x_j + s, \; y_i = y_j \\ -\dfrac{a(x_i, y_i)}{2s} & \text{if } x_i = x_j - s, \; y_i = y_j \end{cases} \tag{9}$$

$$b(i)(x_j, y_j) = \begin{cases} \dfrac{b(x_i, y_i)}{2s} & \text{if } x_i = x_j, \; y_i = y_j + s \\ -\dfrac{b(x_i, y_i)}{2s} & \text{if } x_i = x_j, \; y_i = y_j - s \end{cases} \tag{10}$$

$$c(i)(x_j, y_j) = \frac{c}{s^2} \tag{11}$$

.

Then we replace the gradient w.r.t time in (8) with temporal discretization:

$$\begin{cases} f(t + 1) = \Delta t(a(i)(\nabla f)_{\hat{x}} + b(i)(\nabla f)_{\hat{y}} + c(i)((\Delta f)_{\hat{x}\hat{x}} + (\Delta f)_{\hat{y}\hat{y}})) + f(t) \\ f_i(0) = f_o(i) \end{cases} \tag{12}$$

, where $\Delta t$ is the time step in temporal discretization.

Equation (12) is used for simulating the dynamics described by the equation (5). Then, we uniformly sample 250 points in the above 2D space and choose their corresponding time series of $u$ as the dataset used in our synthetic experiments. We generate 1000 sessions on a $50 \times 50$ regular mesh with time step size $\Delta t = 0.01$. 700 sessions are used for training, 150 for validation and 150 for test.

## A.2 EXPERIMENT SETTINGS

Here we provide additional details for models we used in this work, including model architecture settings and hyper-parameter settings.

### A.2.1 MODEL SETTINGS

Unless mentioned otherwise, all models use a hidden dimension of size 64.

- **VAR**: A vector autoregression model with 2 lags. Input is the concatenated features of previous 2 frames. The weights are shared among all nodes in the graph.
- **MLP**: A multilayer perceptron model with 2 hidden layers. Input is the concatenated features of previous 2 frames. The weights are shared among all nodes in the graph.
- **GRU**: A Gated Recurrent Unit network with 2 hidden layers. Input is the concatenated features of previous 2 frames. The weights are shared among all nodes in the graph.
- **RGN**: A recurrent graph neural network model with 2 GN blocks. Each GN block has an edge update block and a node update block, both of which use a 2-layer GRU cell as the update function. We set its hidden dimension to 73 so that it has the same number of learnable parameters as our proposed model PA-DGN.
- **RGN(StandardOP)**: Similar to RGN, but use the output of difference operators in Section 2.1 as extra input features. We set its hidden dimension to 73.
- **RGN(MeshOP)**: Similar to RGN(StandardOP), but the extra input features are calculated using opeartors in Section 2.2. We set its hidden dimension to 73.
- **PA-DGN**: Our proposed model. The spatial derivative layer uses a message passing neural network (MPNN) with 2 GN blocks using 2-layer MLPs as update functions. The forward network part uses a recurrent graph neural network with 2 recurrent GN blocks using 2-layer GRU cells as update functions.

The numbers of learnable parameters of all models are listed as follows:

Table 4: Numbers of learnable parameters.

| Model | VAR | MLP | GRU | RGN | RGN(StandardOP) | RGN(MeshOP) | PA-DGN |
|---|---|---|---|---|---|---|---|
| **# Params** | 3 | 4,417 | 37,889 | 345,876 | 341,057 | 342,152 | 340,001 |

### A.2.2 TRAINING SETTINGS

**The number of evaluation runs**   We performed 3 times for every experiment in this paper to report the mean and standard deviations.

**Length of prediction**   For experiments on synthetic data, all models take first 5 frames as input and predict the following 15 frames. For experiments on NOAA datasets, all models take first 12 frames as input and predict the following 12 frames.

**Training hyper-parameters**   We use Adam optimizer with learning rate 1e-3, batch size 8, and weight decay of 5e-4. All experiments are trained for a maximum of 2000 epochs with early stopping. All experiments are trained using inverse sigmoid scheduled sampling with the coefficient $k = 107$.

**Environments**   All experiments are implemented with Python3.6 and PyTorch 1.1.0, and are run with NVIDIA GTX 1080 Ti GPUs.

## A.3 EFFECT OF DIFFERENT GRAPH STRUCTURES

In this section, we evaluate the effect of 2 different graph structures on baselines and our models: (1) **k-NN**: a graph constructed with $k$-NN algorithm ($k = 4$); (2) **TriMesh**: a graph generated with Delaunay Triangulation. All graphs use the Euclidean distance.

Table 5: Mean absolute error ($10^{-2}$) for graph signal prediction on the synthetic dataset.

| VAR | MLP | StandardOP | | MeshOP | | SDL | |
|---|---|---|---|---|---|---|---|
| | | $k$-NN | TriMesh | $k$-NN | TriMesh | $k$-NN | TriMesh |
| 17.30 | 16.27 | 12.00 | 12.29 | 12.87 | 12.82 | **11.04** | 12.40 |

Table 6: Graph signal prediction results (MAE) on multistep predictions. In each row, we report the average with standard deviations from all baselines and PA-DGN. One step is 1 hour time interval.

| Region | Method | Graph | 1-step | 6-step | 12-step |
|---|---|---|---|---|---|
| West | VAR | - | $0.1241 \pm 0.0234$ | $0.4295 \pm 0.1004$ | $0.4820 \pm 0.1298$ |
| | MLP | - | $0.1040 \pm 0.0003$ | $0.3742 \pm 0.0238$ | $0.4998 \pm 0.0637$ |
| | GRU | - | $0.0913 \pm 0.0047$ | $0.1871 \pm 0.0102$ | $0.2707 \pm 0.0006$ |
| | RGN | $k$-NN | $0.0871 \pm 0.0033$ | $0.1708 \pm 0.0024$ | $0.2666 \pm 0.0252$ |
| | | TriMesh | $0.0897 \pm 0.0030$ | $0.1723 \pm 0.0116$ | $0.2800 \pm 0.0414$ |
| | RGN (StandardOP) | $k$-NN | $0.0860 \pm 0.0018$ | $0.1674 \pm 0.0019$ | $0.2504 \pm 0.0107$ |
| | | TriMesh | $0.0842 \pm 0.0011$ | $0.1715 \pm 0.0027$ | $0.2517 \pm 0.0369$ |
| | RGN (MeshOP) | $k$-NN | $0.0840 \pm 0.0015$ | $0.2119 \pm 0.0018$ | $0.4305 \pm 0.0177$ |
| | | TriMesh | $0.0846 \pm 0.0017$ | $0.2090 \pm 0.0077$ | $0.4051 \pm 0.0457$ |
| | PA-DGN | $k$-NN | **$0.0840 \pm 0.0004$** | $0.1614 \pm 0.0042$ | **$0.2439 \pm 0.0163$** |
| | | TriMesh | $0.0849 \pm 0.0012$ | **$0.1610 \pm 0.0029$** | $0.2473 \pm 0.0162$ |
| SouthEast | VAR | - | $0.0889 \pm 0.0025$ | $0.2250 \pm 0.0013$ | $0.3062 \pm 0.0032$ |
| | MLP | - | $0.0722 \pm 0.0012$ | $0.1797 \pm 0.0086$ | $0.2514 \pm 0.0154$ |
| | GRU | - | $0.0751 \pm 0.0037$ | $0.1724 \pm 0.0130$ | $0.2446 \pm 0.0241$ |
| | RGN | $k$-NN | $0.0790 \pm 0.0113$ | $0.1815 \pm 0.0239$ | $0.2548 \pm 0.0210$ |
| | | TriMesh | $0.0932 \pm 0.0105$ | $0.2076 \pm 0.0200$ | $0.2854 \pm 0.0211$ |
| | RGN (StandardOP) | $k$-NN | $0.0942 \pm 0.0121$ | $0.2135 \pm 0.0187$ | $0.2902 \pm 0.0348$ |
| | | TriMesh | $0.0868 \pm 0.0132$ | $0.1885 \pm 0.0305$ | $0.2568 \pm 0.0328$ |
| | RGN (MeshOP) | $k$-NN | $0.0913 \pm 0.0016$ | $0.2069 \pm 0.0031$ | $0.2649 \pm 0.0092$ |
| | | TriMesh | $0.0877 \pm 0.0020$ | $0.2043 \pm 0.0026$ | $0.2579 \pm 0.0057$ |
| | PA-DGN | $k$-NN | **$0.0721 \pm 0.0002$** | **$0.1664 \pm 0.0011$** | **$0.2408 \pm 0.0056$** |
| | | TriMesh | $0.0876 \pm 0.0096$ | $0.2002 \pm 0.0163$ | $0.2623 \pm 0.0180$ |

Table 5 and Table 6 show the effect of different graph structures on the synthetic dataset used in Section 3.2 and the real-world dataset in Section 4.2 separately. We find that for different models the effect of graph structures is not homogeneous. For RGN and PA-DGN, $k$-NN graph is more beneficial to the prediction performance than TriMesh graph, because these two models rely more on neighboring information and a $k$-NN graph incorporates it better than a Delaunay Triangulation graph. However, switching from TriMesh graph to $k$-NN graph is harmful to the prediction accuracy of RGN(MeshOP) since Delaunay Triangulation is a well-defined method for generating triangulated mesh in contrast to $k$-NN graphs. Given the various effect of graph structures on different models, our proposed PA-DGN under $k$-NN graphs always outperforms other baselines using any graph structure.

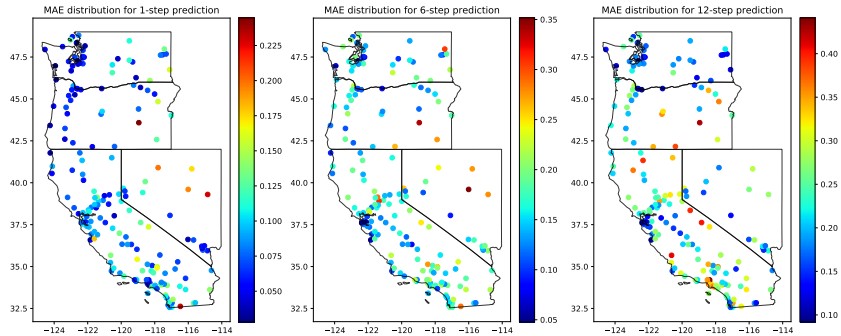

Figure 7: MAE across the nodes.

## A.4 THE DISTRIBUTION OF PREDICTION ERROR ACROSS NODES

Figure 7 provides the distribution of MAEs across the nodes of PA-DGN applied to the graph signal prediction task of the west coast region of the real-world dataset in Section 4.2. As shown in the figure, nodes with the highest prediction error for short-term prediction are gathered in the inner part where the observable nodes are sparse, while for long-term prediction nodes in the area with a limited number of observable points no longer have the largest MAE. This implies that PA-DGN can utilize neighboring information efficiently even under the limitation of sparsely observable points.

## A.5 EVALUATION ON NEMO SEA SURFACE TEMPERATURE (SST) DATASET

We tested our proposed method and baselines on the NEMO sea surface temperature (SST) dataset[1]. We first download the data in the area between $50N^\circ$-$65N^\circ$ and $75W^\circ$-$10W^\circ$ starting from 2016-01-01 to 2017-12-31, then we crop the $[0, 550] \times [100, 650]$ square from the area and sample 250 points from the square as our chosen dataset. We divide the data into 24 sequences, each lasting 30 days, and truncate the tail. All models use the first 5-day SST as input and predict the SST in the following 15 and 25 days. We use the data in 2016 for training all models and the left for testing.

For StandardOP, MeshOP and SDL, we test both options using linear regression and using RGN for the prediction part and report the best result. The results in Table 7 show that all methods incorporating spatial differences gain improvement on prediction and that our proposed learnable SDL outperforms all other baselines.

Table 7: Mean absolute error ($10^{-2}$) for SST graph signal prediction.

|         | VAR    | MLP    | GRU    | RGU    | StandardOP | MeshOP | SDL    |
|---------|--------|--------|--------|--------|------------|--------|--------|
| 15-step | 15.123 | 15.058 | 15.101 | 15.172 | 14.756     | 14.607 | **14.382** |
| 25-step | 19.533 | 19.473 | 19.522 | 19.705 | 18.983     | 18.977 | **18.434** |

---

[1]Available at http://marine.copernicus.eu/services-portfolio/access-to-products/?option=com_csw&view=details&product_id=GLOBAL_ANALYSIS_FORECAST_PHY_001_024.

## A.6 EVALUATION ON DATASETS WITH DIFFERENT SPARSITY

We changed the number of nodes to control the sparsity of data. As shown in Table 8, our proposed model outperforms others under various settings of sparsity on the synthetic experiment in Section 3.2.

Table 8: Mean absolute error $(10^{-2})$ for graph signal prediction with different sparsity.

| #Nodes | VAR | MLP | StandardOP | MeshOP | SDL |
|--------|--------|--------|------------|--------|------------|
| 250 | 0.1730 | 0.1627 | 0.1200 | 0.1287 | **0.1104** |
| 150 | 0.1868 | 0.1729 | 0.1495 | 0.1576 | **0.1482** |
| 100 | 0.1723 | 0.1589 | 0.1629 | 0.1696 | **0.1465** |

Furthermore, we sampled 400 points and trained SDL as described in Section 3.1, and resampled fewer points (350,300,250,200) to evaluate if SDL generalizes less sparse setting. As Table 9 shows, MSE increases when fewer sample points are used. However, SDL is able to provide much more accurate gradients even if it is trained under a new graph with different properties. Thus, the results support that SDL is able to generalize the c setting.

Table 9: Mean squared error $(10^{-2})$ for approximations of directional derivatives of function $f_2(x, y) = \sin(x) + \cos(y)$ with different sparsity.

| Method | 350 Nodes | 300 Nodes | 250 Nodes | 200 Nodes |
|--------|-----------------|-----------------|-----------------|-----------------|
| FinGrad | $2.88 \pm 0.11$ | $3.42 \pm 0.14$ | $3.96 \pm 0.17$ | $4.99 \pm 0.31$ |
| SDL | $\mathbf{1.03 \pm 0.09}$ | $\mathbf{1.14 \pm 0.12}$ | $\mathbf{1.40 \pm 0.10}$ | $\mathbf{1.76 \pm 0.10}$ |

