# OpenReview forum: "Physics-aware Difference Graph Networks for Sparsely-Observed Dynamics"
_ICLR.cc/2020/Conference — Accept (Poster)_

### Official Review · AnonReviewer2 · 2019-10-23
**Official Blind Review #2**

**Rating:** 6

**Review:**

This paper proposes a method to reduce numerical error when predicting sequences governed by physical dynamics. The idea is that most physics simulators use finite difference differential operators, and the paper adds trainable parameters to them (the derivative and Laplacian operators). The added parameters are computed by a GNN. The paper concats the original graph feature and the output of the differential operators, and inputs them to a recurrent GNN to obtain the final prediction.

I think the idea is interesting. Incorporating modulated derivative and Laplacian operators into physical simulators is novel and well justified. It could strengthen the argument is there is more justification of why this particular parameterization is selected.

I think the experimental evaluation is somewhat adequate. There are a good selection of baselines including both manually designed iterators and GNNs. In particular, the weather prediction experiment show improved performance over several baselines. I am not familiar with this task or its state-of-the-art performance, but I am convinced that the proposed approach is superior compared to the claimed baselines (RGN, GRU).

I have several confusions or concerns about the synthetic experiments

1. In the synthetic experiments, is the evaluation task different from the training task? It is unclear from the description how well the learned parameters generalize. Does the method generalize to a. New functions/dynamics b. New graphs with similar properties (e.g. another graph draw from the same distribution) c. New graph with different properties (e.g. more or less sparse)?

2. One short-coming of the synthetic experiment is the lack of error bars, or analysis of statistical significance. I think some of the improvements are not large enough to be statistically convincing without additional analysis. It seems necessary to experiment on multiple random problems (e.g. with random meshing, dynamics parameters).

Minor comments:

A related idea is “Learning Neural PDE Solvers with Guarantees” which modulates the finite difference iterative solver with deep networks, but the objective is solving PDEs with known dynamics instead of prediction with unknown dynamics. Conversely, the method the authors proposed seem also useful for speeding up PDE solvers.

I think there is an error in the type definition of f and F in section 2.1. The two claimed types contradict each other.


**Experience Assessment:**

I have published one or two papers in this area.

**Review Assessment: Checking Correctness Of Derivations And Theory:**

I assessed the sensibility of the derivations and theory.

**Review Assessment: Checking Correctness Of Experiments:**

I assessed the sensibility of the experiments.

**Review Assessment: Thoroughness In Paper Reading:**

I read the paper at least twice and used my best judgement in assessing the paper.

---

> ### Author Response · Authors · 2019-11-12
> **Response #2**
>
> Thank you for your comments and suggestions to improve the paper. Below are our responses to the main points of your comments:
>
> >> “Justification of why this particular parameterization is selected”
> Thanks for pointing out the motivation of the form, Eq1. As we mentioned in the draft, the main idea of SDL is to provide “modulated gradients and Laplacian” which are more effective approximations of the derivatives than the approximation of finite derivatives. Since there are two variables (f_i and f_j) involved in the gradient and Laplacian, it is natural to introduce two learnable parameters. In fact, Eq 1 is a form of affine transform (excluding bias term). The reason why we didn’t follow w1*f_j + w2*f_i, which is more generic, is to distinguish the role of each term. In other words, w^(1) is a scaling term and w^(2) is a differencing term. By doing that, we can enforce some constraints to the learnable parameters (e.g., make 0<w<1 or w is positive, etc.) separately for some purpose and it is easier to see how the constraints affect the derivatives.
>
>
> Several confusions or concerns about the synthetic experiments:
> >> “1. Evaluation task and training task”
> The synthetic experiments (3.1 and 3.2) are under the supervised setting and therefore, the evaluation and training tasks are the same.
>
> >> “1. What does the method generalize?”
> In terms of the generalization, the SDL generalizes the b setting, New graphs with a similar number of different sampled points. In other words, the method can learn parameters to compute the derivatives from discrete samples and the parameters are still valid for the different but same number of sample points. This generalization is verified by our synthetic experiments in Section 3.1.
>
> Furthermore, if the number of samples is enough, it also generalizes the c setting, New graph with different properties (e.g. more or less sparse).
> We sampled 400 points and trained SDL as described in Section 3.1, and resampled fewer points (350,300,250,200) to evaluate if SDL generalizes less sparse setting. As the following table shows, MSE increases when fewer sample points are used. However, SDL is able to provide much more accurate gradients even if it is trained under a new graph with different properties. Thus, the results support that SDL is able to generalize the c setting.
>
> Mean squared error (10^−2) for approximations of directional derivatives.
> +---------------+-------------------+---------------+-------------------+---------------+
> | Functions | FinGrad(350) | SDL(350)  | FinGrad(300) | SDL(300)  |
> | f2(x,y)       |    2.88±0.11    |  1.03±0.09 |    3.42±0.14    | 1.14±0.12 |
> +---------------+-------------------+---------------+-------------------+---------------+
> |                    | FinGrad(250)|  SDL(250)  | FinGrad(200) | SDL(200)  |
> |                    |   3.96±0.17    |  1.40±0.10 |     4.99±0.31    | 1.76±0.10 |
> +---------------+-------------------+---------------+-------------------+---------------+
>
> The parameters consider the function values at each sampled point and spatial displacement between two points, thus if the dynamics/functions are changed, our parameters won’t be applicable without training on the new dataset.
>
> >> “2. Error bars on the synthetic experiment”
> We provide the standard deviation for the synthetic experiments in Section 3.1.
>
> Table 1: Mean squared error (10^−2) for approximations of directional derivatives.
> +---------------+--------------+---------------+--------------+---------------+--------------+
> | Functions |  FinGrad  |      MLP      |       GN      |   One-w    |      SDL      |
> |  f1(x,y)      | 6.42±0.47 | 2.12±0.32 | 1.05±0.42 | 1.41±0.44 | 0.97±0.39 |
> |  f2(x,y)      | 5.90±0.04 | 2.29±0.77 | 2.17±0.34 | 6.73±1.17 | 1.26±0.05 |
> +---------------+--------------+---------------+--------------+---------------+--------------+
>
> We provide the standard deviation for the synthetic experiments in Section 3.2.
> We generated 3 random meshes for the synthetic experiment in Section 3.2 and reported the mean absolute errors of all methods on the graph signal prediction task. Results show that our proposed method outperforms baselines significantly.
>
> Table 2: Mean absolute error (10^−2) for graph signal prediction
> +----------------+----------------+------------------+----------------+----------------+
> |       VAR       |      MLP       | StandardOP |   MeshOP   |      SDL       |
> | 16.84±0.41 | 15.75±0.53 |  11.90±0.29  | 12.82±0.06 | 10.87±0.98 |
> +----------------+----------------+------------------+----------------+----------------+
>
>
> Minor comments:
> >> “A related idea”
> This is a great suggestion. Actually, we are going to work for the applications of SDL and this point will be a possible future direction.
>
> >> “the type definition of f and F”
> In Section 2.1, we define f as node feature and F as edge feature. While we believe that the definition is correctly described, if you could point out the contradiction, we will make it clear.

---

### Official Review · AnonReviewer3 · 2019-10-23
**Official Blind Review #3**

**Rating:** 8

**Review:**

The authors propose new architectures that simulate difference operators. According to my experience, this research is important since PDEs are the most commonly used form to represent known physical relationships in dynamical systems.  The proposed method has novelty in that it uses advanced machine learning architectures to estimate physically meaningful variables. The authors further investigate how to use the proposed spatial difference layer in two tasks.
I would suggest improving this research on these aspects:
1.	The proposed method should be evaluated on more datasets. The difference information is used in almost any real-world dynamical systems and thus it would be more convincing to show the effectiveness on diverse applications, e.g., object tracking, the variation of energy and mass across space and time.
2.	It would be interesting to design a test scenario where governing PDEs are known. Is it possible for your method to uncover the relationship of gradients that govern the system?
3.	How sparse the data is? It would be better to have an experiment where data is intentionally hidden to control the sparsity and then evaluate the performance.
4.	A side question: In real-world systems, the observations are not only governed by PDE, but also unknown noisy factors, missing physics, etc. Can your method handle the noisy data/labels?
5.	The proposed method has some complex components. I would encourage releasing the code and the simulated dataset upon acceptance.


**Experience Assessment:**

I have published one or two papers in this area.

**Review Assessment: Checking Correctness Of Derivations And Theory:**

I assessed the sensibility of the derivations and theory.

**Review Assessment: Checking Correctness Of Experiments:**

I carefully checked the experiments.

**Review Assessment: Thoroughness In Paper Reading:**

I read the paper at least twice and used my best judgement in assessing the paper.

---

> ### Author Response · Authors · 2019-11-12
> **Response #3**
>
> Thank you for your comments and suggestions to improve the paper. Below are our responses to the main points of your comments:
>
> >> “1. Evaluation on More datasets”
> Thanks for suggesting the evaluation on more datasets. The proposed method assumes that there are continuous phenomena governed by physics rules or equations and the phenomena are only observed at some locations (i.e., sparsely-observed dynamics). Under this assumption, we thought that climate data is ideal to evaluate our idea.
> Currently, we are looking for more datasets to support our idea and we are investigating to apply on Sea Surface Temperature dataset.
>
> >> “2. Uncover the relationship of gradients”
> This is a great suggestion and actually, discovering (or uncovering) hidden physics is one of the research topics related to physics with deep learning. In fact, while the main motivation of our work is different to these kinds of work, we believe that it is a great idea to extend our work to discovering latent rules since it makes data-driven models more interpretable.
>
> >> "3. How sparse the data is?"
> For the synthetic experiment in Section 3.1, we sampled 200 points in 2D space (x,y)∈[-5.0, 5.0] x [-5.0, 5.0].
> For the synthetic experiment in Section 3.2, we sampled 250 points in 2D space (x,y)∈[0, 2π] x [0, 2π].
> For the temperature prediction experiment in Section 4,
> +---------------+----------------+---------------------+
> |                    |    Western  | Southeastern  |
> |   # Nodes  |        191       |         230            |
> +---------------+----------------+---------------------+
>
> >> "3. Controlling the sparsity and evaluation the performance"
> We changed the number of nodes to control the sparsity of data. Our proposed model outperforms others under various settings of sparsity on the synthetic experiment in Section 3.2.
>
> Mean absolute error (10^−2) for graph signal prediction
> +---------------+-----------+-----------+------------------+-------------+------------+
> |     Graph    |    VAR   |   MLP   | StandardOP | MeshOP |   Ours   |
> | 250 nodes | 0.1730 | 0.1627 |       0.1200     |    0.1287  |  0.1104  |
> | 150 nodes | 0.1868 | 0.1729 |       0.1495     |    0.1576  |  0.1482  |
> | 100 nodes | 0.1723 | 0.1589 |       0.1629     |    0.1696  |  0.1465  |
> +---------------+-----------+-----------+------------------+-------------+------------+
>
> Furthermore, we sampled 400 points and trained SDL as described in Section 3.1, and resampled fewer points (350,300,250,200) to evaluate if SDL generalizes less sparse setting. As the following table shows, MSE increases when fewer sample points are used. However, SDL is able to provide much more accurate gradients even if it is trained under a new graph with different properties. Thus, the results support that SDL is able to generalize the c setting.
>
> Mean squared error (10^−2) for approximations of directional derivatives.
> +---------------+-------------------+---------------+-------------------+---------------+
> | Functions | FinGrad(350) | SDL(350)  | FinGrad(300) | SDL(300)  |
> | f2(x,y)       |    2.88±0.11    |  1.03±0.09 |    3.42±0.14    | 1.14±0.12 |
> +---------------+-------------------+---------------+-------------------+---------------+
> |                    | FinGrad(250)|  SDL(250)  | FinGrad(200) | SDL(200)  |
> |                    |   3.96±0.17    |  1.40±0.10 |     4.99±0.31    | 1.76±0.10 |
> +---------------+-------------------+---------------+-------------------+---------------+
>
> >> “4. Can your method handle the noisy data/labels?”
> In this work, we assume that SDL is able to learn more effective approximations of derivatives, which are essential elements in physics dynamics, and some noisy factors can be handled by data-driven learning similar to many deep models. We choose graph neural networks for prediction in temperature and the learnable parameters handle the noisy factors and missing physics.
>
> >> “5. Releasing the code and dataset”
> We will release the code and the dataset upon acceptance.

---

> > ### Author Response · Authors · 2019-11-14
> > **“1. Evaluation on More datasets”**
> >
> > We tested our proposed method and baselines on the NEMO sea surface temperature (SST) dataset (available at http://marine.copernicus.eu/services-portfolio/access-to-products/?option=com_csw&view=details&product_id=GLOBAL_ANALYSIS_FORECAST_PHY_001_024)
> > We first download the data in the area between 50N-65N and 75W-10W starting from 2016-01-01 to 2017-12-31, then we crop the [0, 550] - [100, 650] square from the area and sample 250 points from the square as our chosen dataset. We divide the data into 24 sequences, each lasting 30 days, and truncate the tail. All models use the first 5-day SST as input and predict the SST in the following 15 and 25 days. We use the data in 2016 for training all models and the left for testing.
> >
> > For StandardOP, MeshOP and SDL, we test both options using linear regression and using RGN for the prediction part and report the best result. The results show that all methods incorporating spatial differences gain improvement on prediction and that our proposed learnable SDL outperforms all other baselines.
> >
> > Mean absolute error (10^−2) for SST prediction
> > +--------+-----------+----------+----------+-----------+------------------+-------------+-----------+
> > | Step |    VAR   |  MLP    |   GRU  |   RGN   | StandardOP | MeshOP |   SDL    |
> > |  15    | 15.123 | 15.058 | 15.101 | 15.172 |     14.756        |   14.607  | 14.382 |
> > |  25    | 19.533 | 19.473 | 19.522 | 19.705 |     18.983        |   18.977  | 18.434 |
> > +--------+-----------+----------+----------+-----------+------------------+-------------+-----------+

---

### Official Review · AnonReviewer1 · 2019-10-24
**Official Blind Review #1**

**Rating:** 8

**Review:**

Summary:
The paper considers the problem of predicting node and/or edge attributes in a physics-governed system, where we have access to measurement data on discrete sensor locations. In contrast to other methods, they explore the usefulness of a Spatial Difference Layer (SDL) that learns a representation of the gradients on the edges and the Laplacian on the nodes, where the parameters to create those operators are learnable weights. The SDL layer is concatenated with the original graph and fed into a recurrent graph network (RGN) to predict node and/or edge properties.

Strengths:
- This research is very relevant for physics inspired machine learning, since many physical systems are governed by underlying differential equations.
- The authors show on synthetic data experiments that SDL is capable of representing the derivatives of a physical system.
- Real world use case for temperature prediction is presented with encouraging results.

Weaknesses:
- While comparison to RGN represents a rather strong benchmark, it would be interesting to see a comparison to a graph learning model that is specifically designed for weather forecast.
- Just adding the Spatial Difference Layer using numerical methods (method RGN(StardardOP) and RGN(MeshOP)) can diminish prediction power for a long time horizon. This result suggests that those gradients might not help the prediction.
- The inclusion of an h-hop neighborhood is not quite clear. What value for h was used in the experiments? Is this really necessary, when RGN by itself propagates the signal to neighbors that are further away?

Additional comments:
- 2.1 and 2.2: On first reading, it's a bit confusing why there are two different equations for (∆f)i. The motivation of the second equation should be made more explicit.
- 3.1 lacks explanation of what is train / test set, which is given only in the appendix. This is critical information to understand the use cases of the model and should definitely be in the main body.
- In 3.2 formatting of a(i), b(i) and c(i) is confusing. Why is the setup only similar to Long et al? It would be nice to point out the differences and explain why it wasn't exactly the same.
- 4.1: Was the train/validation/test split done in contiguous segments? I.e. are the 8 months of training data January to August? How is the problem of learning different seasons handled?


**Experience Assessment:**

I have read many papers in this area.

**Review Assessment: Checking Correctness Of Derivations And Theory:**

I assessed the sensibility of the derivations and theory.

**Review Assessment: Checking Correctness Of Experiments:**

I assessed the sensibility of the experiments.

**Review Assessment: Thoroughness In Paper Reading:**

I read the paper at least twice and used my best judgement in assessing the paper.

---

> ### Author Response · Authors · 2019-11-12
> **Response #1**
>
> Thank you for your comments and suggestions to improve the paper. Below are our responses to the main points of your comments:
>
> Weaknesses:
> >> “A comparison to a graph learning model that is specifically designed for weather forecast”
> It would be a great comparison if there is a graph-based model specifically designed for a weather forecast. However, to the best of our knowledge, we haven’t found the existing graph-based model particularly designed for climate modeling under the sparsely-observed setting. If you are aware of any references about a graph learning model for weather forecasting, please let us know them.
>
> >> “Adding StandardOP or MeshOP might not help the prediction.”
> It is a good point that adding not-learnable operators (StandardOP or MeshOP) doesn’t help to reduce the prediction error. The inconsistent behaviors of RGN(StandardOP) and RGN(MeshOP) in Table 3 support the idea that incorporating some incorrect or irrelevant features is little helpful and it can even be harmful. As Section 3.1 shows, the operators having no learnable parameters suffer substantial numerical error under the sparse setting and it causes the diminished prediction power when the operators are used. On the other hand, the consistent prediction power from PA-DGN is the evidence that the physics-inspired features from SDL are significantly helpful and provide a more effective inductive bias.
>
> >> “What value for h was used?”
> Thanks for pointing out the confusion. Overall, there are two GNNs involved in PA-DGN; (1) GNNs in SDL and (2) GNNs in RGN. For both GNNs, we used h=2.
>
> >> “Necessity of SDL since RGN by itself propagates the signals to neighbors”
> This is a very good point. Yes, RGN is actually able to do message-passing and it means that it is able to incorporate neighboring features. However, as the expressive power and learning efficiency of data-driven models are highly dependent on its architecture and features extracted from itself, it is still very critical to design proper architecture for efficient learning.
> The purpose of using SDL is to provide physics-aware representations, which are data-driven spatial derivatives (gradients and Laplacian), instead of using observations directly. In other words, SDL provides a physics-aware inductive bias, which improves the prediction quality.
>
>
> Additional comments:
> >> “The motivation of the second (∆f)i”
> The second (∆f)i in Section 2.2 is introduced to provide a different form of Laplacian in a triangulated mesh. It is well-known that the second Laplacian (geometric discretizations
> of the Laplacian) has more effective approximation qualities on the mesh. We will update the motivation of the second one in the draft.
>
> >> “Lack of explanation of what is train / test set”
> For the experiment in Section 3.1, we first defined a function on 2D space. Then, we sampled 200 points from the 2D coordinates and built a graph based on k-NN algorithm. As gradients are defined on each edge, we split the available edges as train/validation/test sets. We will update the draft to provide this information clearly.
>
> >> “Why the setup in Section 3.2 is only similar to PDE-Net?”
> There are 2 differences between the settings of Equation (8) in Long et al. 2018 and ours. (1) While the coefficients in Long et al. 2018 before the second-order spatial differentiation terms in the partial differential equation are constant, they are from a function of the coordinates of nodes in our setting. This setting increases the dynamics of the generated datasets and makes the prediction task more challenging; (2) In Long et al. 2018, the second-order spatial differentiation terms along x-axis and y-axis have different coefficients, while in ours setting they share the same coefficients. We make this modification to fit the equation on graph-structured data, because the Laplacian term on graphs of sampled nodes is defined as a scalar on nodes instead of having a specific direction.
>
> >> “Data split and the problem of learning different seasons”
> Thanks for pointing out the data splitting. Yes, we used the first 8 months for training and left months for validation and test. In fact, learning different seasons doesn't matter much since we focus on "differences" instead of absolute values. In other words, our model is focusing on how the "differences" of physical quantities interact and propagate spatially and temporally, and thus, if the governing physics rules are not significantly changed over the different seasons, it won’t be affected.
> Still, some unique characteristics in the specific months in test/validation sets can’t be seen during the training and they may not be properly handled. While we only have one-year observations, this problem can be handled by yearly splitting, i.e., train a model using a certain year and evaluate it on another year.

---

### Decision · Program_Chairs · 2019-12-19

**Decision:**

Accept (Poster)

**Comment:**

All reviewers agree that this research is novel and well carried out, so this is a clear accept. Please ensure that the final version reflect the reviewer comments and the new information provided during the rebuttal